# Chasing the Target: New Phenomena of Resistance to Novel Selective RET Inhibitors in Lung Cancer. Updated Evidence and Future Perspectives

**DOI:** 10.3390/cancers13051091

**Published:** 2021-03-04

**Authors:** Sara Fancelli, Enrico Caliman, Francesca Mazzoni, Marco Brugia, Francesca Castiglione, Luca Voltolini, Serena Pillozzi, Lorenzo Antonuzzo

**Affiliations:** 1Medical Oncology Unit, Careggi University Hospital, 50134 Florence, Italy; sara.fancelli@unifi.it (S.F.); enrico.caliman@unifi.it (E.C.); mazzonifr@aou-careggi.toscana.it (F.M.); brugiam@aou-careggi.toscana.it (M.B.); serena.pillozzi@unifi.it (S.P.); 2Department of Experimental and Clinical Medicine, University of Florence, 50134 Florence, Italy; luca.voltolini@unifi.it; 3Pathological Histology and Molecular Diagnostics Unit, Careggi University Hospital, 50134 Florence, Italy; castiglionef@aou-careggi.toscana.it; 4Thoraco-Pulmonary Surgery Unit, Careggi University Hospital, 50134 Florence, Italy

**Keywords:** RET, NSCLC, selpercatinib, pralsetinib, solvent-front mutations, acquired resistances

## Abstract

**Simple Summary:**

REarranged during Transfection (RET) is an emerging target for several types of cancer, including non-small cell lung cancer (NSCLC). The recent U.S. FDA approval of pralsetinib and selpercatinib raises issues regarding the emergence of secondary mutations and amplifications involved in parallel signaling pathways and receptors, liable for resistance mechanisms. The aim of this review is to explore recent knowledge on RET resistance in NSCLC in pre-clinic and in clinical settings and accordingly, the state-of-the-art in new drugs or combination of drugs development.

**Abstract:**

The potent, RET-selective tyrosine kinase inhibitors (TKIs) pralsetinib and selpercatinib, are effective against the RET V804L/M gatekeeper mutants, however, adaptive mutations that cause resistance at the solvent front RET G810 residue have been found, pointing to the need for the development of the next-generation of RET-specific TKIs. Also, as seen in EGFR- and ALK-driven NSCLC, the rising of the co-occurring amplifications of KRAS and MET could represent other escaping mechanisms from direct inhibition. In this review, we summarize actual knowledge on RET fusions, focusing on those involved in NSCLC, the results of main clinical trials of approved RET-inhibition drugs, with particular attention on recent published results of selective TKIs, and finally, pre-clinical evidence regarding resistance mechanisms and suggestion on hypothetical and feasible drugs combinations and strategies viable in the near future.

## 1. Introduction

Recent evidence in non-small cell lung cancer (NSCLC) about the new highly selective REarranged during Transfection (RET) inhibitors selpercatinib and pralsetinib, despite impressive results in clinical trials, have raised the urgency to highlight the best therapeutic sequence in view of novel resistances which are able to cause selective inhibition to be useless. With this aim, and aware that the inhibition of RET has always been a challenge since its discovery in T-cell lymphoma [1], we want to summarize the available knowledge on RET inhibitors, including both ongoing trials with new drugs and pre-clinical data concerning overcoming the resistance mechanisms.

## 2. RET in Pills

*RET* is located on chromosome 10q11.2 and its expression is mediated by several DNA-binding proteins belonging to the Sp family of transcription factors (Sp1, Sp3) [2] or early growth response protein 1 (EGR1) [2], SRY-box 10 (SOX10), paired box 3 (PAX3) [3], NK2 homeobox 1 (NKX2–1) and homeobox B5 (HOXB5) [4]. *RET* encodes for a Transmembrane Tyrosine Kinase Receptor (RTK) with a unique structure composed of four cadherin-like domains, a cysteine-rich domain, a transmembrane domain and a tyrosine kinase (TK) domain, this latter has a different number of amino acids depending on the isoform transcribed (RET9, RET43 and RET51) [5]. Each isoform interacts with adaptors and signaling proteins that are able to activate different downstream pathways during embryogenesis, in homeostasis of several tissues [6]. Physiologically, beginning RET signals depend on the binding of specific ligand members of the glial cell line-derived neurotrophic factors (GDNFs) with GDNF family receptor alpha (GFRα). The ligand family includes GDNF, neurturin (NTRN), artemin (ARTN) and persephin (PSPN) and each has a selective, although not completely specific, receptor, respectively called GFRα1, GFRα2, GFRα3 and GFRα4 [7]. The interposition of the GDNF-GFRα complex allows for the homodimerization between RET monomers resulting in autophosphorylation of the intracellular tyrosine residues of the main docking-site of the RET51 isoform (Y1062). RET is also able to heterodimerize with other RTKs [5]. Phosphorylated tyrosine recruits a multitude of adaptors that, in turn, mediate the activation of RAS- Mitogen-Activated Protein Kinases (MAPK) and Phosphatidylinositol-3 Kinase (PI3K)- Protein Kinase B (AKT) pathways [5]. Several docking sites (Y900, Y905, Y981, Y1015 and Y1096), able to trigger additional downstream pathways such as JAK/STAT, PKA, PKC and JNK, have been described [6]. Moreover, RET interacts with RTKs and other cell surface proteins guaranteeing a continuity and spreading of downstream signals [8] (Figure 1). During embryogenesis RET is mainly expressed in the urinary tract, nervous system and hematopoietic stem cells, justifying the pathogenesis of hereditary diseases secondary to germline mutations (loss of function). In adult life, low levels of RET expression are registered in all tissues [9] and different RET molecular alterations have been reported in tumors at either germline or somatic levels. These include gene amplification, fusion, as well as single base substitutions/small insertions/deletions.

### 2.1. Germline Mutations

Genitourinary and nervous system development [10,11], maturation and migration of stem cell lines and a general involvement in embryogenesis and spermatogenesis, represent the main known mechanisms in which RET’s signaling is involved during embryonic development [12,13]. It’s clearly understandable how RET loss of function due to germline mutations, affecting those mechanism, can lead to a variety of congenital malformations such as Hirschsprung disease (HSCR) and congenital abnormalities of the kidney and urinary tract (CAKUT), and cause numerous symptoms in patients with phenotypic variants of these syndromes [14,15]. However, a role for RET in maintenance of hematopoietic system and in development of Gut-Associated Lymphoid Tissue (GALT) has recently been recognized [16]. Germinal mutations of the proto-oncogene RET affecting cysteine-rich extracellular domains or less frequently on the intracellular domains give rise to multiple neuroendocrine neoplasia 2 (MEN2). MEN2 is classified based on clinical features in MEN2A characterized by thyroid cancer, pheochromocytoma, and hyperparathyroidism and in MEN2B with also ganglioneuromatosis and a Marfanoid habitus [17]. Similarly to MEN2, the familial medullary thyroid carcinoma (FMTC) derived from germinal point mutation that causes an increase in the effect of self-activation by increasing ATP-binding or phosphorylation activity, sustains the oncogenic and pro-proliferative stimuli [18]. Every point mutation, rarely seen outside neuroendocrine neoplasms, correlates with different prognosis and clinical outcome, suggesting the necessity to sketch out an early screening and subsequently a different therapeutic approach [19,20]. Indeed, MEN2A and FMTC, having phenotypic and clinical indolent characteristics, appear to be more of a continuum of the same disease, unlike MEN2B which has a juvenile onset and a more aggressive course [21].

### 2.2. Somatic Mutations and Cancer

To better understand its decisive role as a proto-oncogene in sporadic cancers we had to wait until the Chernobyl disaster in 1986 showed a correlation between the papillary thyroid carcinomas (PTC) onset and gene rearrangements in post-radiation exposed children [22]. RET/Coiled-Coil Domain Containing 6 (CCDC6) gene fusion is associated in about 80% of cases of sporadic PTC, while the Nuclear Receptor Coactivator 4 (NCOA4) gene, is mainly related to radiation exposure and younger age [23]. Countless rearrangements have been described in literature being a part of the pathogenesis of PTC [24]. Although PTCs are the most frequently associated cancers. with RET rearrangements (10–20%), many other neoplasms are associated with RET-fusion involved in creating resistances and escaping mechanisms to classical therapies. In hormone positive breast cancer (BC), RET overexpression is described in less than 0.1% of cases and is involved in resistance to anti-hormonal therapies in BC cell lines [25]. Based on preclinical evidence of crosstalk between RET and positive estrogen receptors, some clinical trials in BC patients without any convincing results in disease control explore the benefit of using multi-kinase inhibitors active on RET [26]. Recently, a single case report has been presented as part of LIBRETTO-001 trial, of a metastatic BC woman who presented a complete clinical response with Selpercatinib, suggesting a possible role of selective RET inhibitors in this field [27]. In colon cancers RET rearrangements represent 0.2% of cases [28]. Among them, 2/3 manifest in the right colon and are characterized by MSI, RAS and BRAF wild type status, and could benefit from the use of specific therapies [29,30]. Other gastrointestinal malignancies, gynecological tumors, renal and prostate cancer have a limited expression of RET fusions [31] and could benefit from treatment inside basket trials, thanks to recent gene sequencing techniques.

### 2.3. RET in Lung Cancer

In NSCLC the prevalence of RET alterations is estimated to be 1–2% of all cases [32]. Thanks to modern genomic sequencing methods, the first fusion gene discovery in 2011 between RET and the Kinesin Family Member 5B (KIF5B) gene has allowed to broaden the knowledge of translocations involving RET [33]. As mentioned above, rearrangements involving chromosome 10 are intrachromosomal, leading to fusion with several genes lying on the same chromosome. In NSCLC the gene most involved in fusions is KIF5B, a gene involved in a pericentric rearrangement, followed by CCDC6 and NCOA4 which are characterized by a paracentric inversion fusion [32]. Several other inter-chromosomal rearrangements or translocations have been described, however they represent a small percentage of cases, we have summarized some in Table 1. [24,34]. Breakpoints in KIF5B are frequently found in the intron 11 at different positions and are involved in transcription of intracytoplasmic segments of RET, however, different introns are rarely involved in fostering the inclusion of the transmembrane dominion [35,36]. In addition to fusions, single amplifications or mutations with variable penetrance related to histotype and gender have been found [34,37]. KIF5B exon 15 fusion to RET exon 12 is the most frequently detected fusion in nonsmokers and young females, while CCDC6 exon 1 to RET exon 12 correlates with smoking habits and male gender [34].

Adenocarcinoma and adenosquamous carcinoma are the most frequent histologies diagnosed in rare solid subtypes as per signet ring cells or mucinous form [38]. Fusions support intracellular signaling thanks to the increase in kinase expression in those tissues normally lacking in RET expression, non-ligand mediated self-activation secondary to mutation in upstream proteins able to support coiled-coil domains interaction, and finally, the loss of self-inhibitory capacity. These modifications determine activation of the signaling pathways STAT3, JAK/STAT3 and RAS/RAF/MEK/ERK capable of supporting proliferation, differentiation, angiogenesis and metastasis as demonstrated in in vivo experiences [39,40,41].

## 3. Activity of MKIs in RET-Positive NSCLC

As the tyrosine kinase receptor RET shares similarities in the structure of the kinase domain with other tyrosine kinases (TK) [56], initial attempts to target RET rearrangements focused on multikinase inhibitors (MKIs) with non-selective RET inhibitory activity have been approached. However, results obtained with MKIs suggest that RET fusions are not highly actionable. Treatment with MKIs in RET fusion-positive NSCLC demonstrated both modest clinical activity and limited response durability. Moreover, the response rates achieved in clinical experiences were lower compared with outcomes with therapies targeting other oncogenic drivers (i.e., EGFR mutations, ALK and ROS-1 fusions) [57]. Several MKIs inhibitors, that have been investigated in the treatment of RET-rearranged NSCLC, are approved for the treatment of thyroid cancers (i.e., vandetanib, cabozantinib, lenvatinib and sorafenib) or are approved for other indications (i.e., ponatinib, alectinib and sunitinib). Activity of MKIs in RET fusion-positive NSCLC has been reported in pre-clinical cancer models [31,36,58,59], in retrospective case series [45,60] and in phase I and phase II trials [43,49,54,61,62,63,64]. These agents have been developed against a variety of target-kinases other than (or in addition to) RET, such as VEGF receptors, AXL, FGFR1, EGFR, MET, c-KIT and BRAF, and unfortunately demonstrated limited potency for RET-positive cancers. Moreover, these agents have led to a variety of adverse events (AEs) which are closely related to their activity against other pathways, such as EGFR (diarrhea and dermatologic toxicities) and VEGFR (hypertension). These off-target side effects can frequently lead to discontinuation of treatment or dose reduction and, as a consequence, treatment with a dose that effectively inhibits RET would not be guaranteed. Taken together these evidence may explain the suboptimal activity and the lower clinical benefits obtained in MKIs-treated RET-positive NSCLC, compared to the outcomes of other oncogene-addicted NSCLC subtypes when treated with matched targeted therapies. 

### 3.1. Vandetanib

Vandetanib, a multi-target TKI targeting VEGF receptors, EGFR and RET, has been investigated in a phase II trial [61]. In this study clinical antitumor activity has been reported in nine out 19 (47%) patients with RET-positive NSCLC enrolled (ORR 47%), but grade 3 or 4 AEs were common and a dose reduction was described in 53% of patients. In another phase II trial [49] vandetanib showed moderate activity in pretreated patients with NSCLC harboring RET rearrangements (ORR 18%, DCR 65%) and dose reduction was necessary in 4 of 18 (22%) patients enrolled. The efficacy outcomes reported in the above studies are comparable with the retrospective analysis of a global registry study (GLORY) with an ORR = 18% [45]. Moreover, in a retrospective analysis of four randomized phase III trials, the overall prevalence of RET rearrangements identified was 0.7% and none of the three RET-positive NSCLC patients have obtained an OR after treatment with vandetanib [65]. 

### 3.2. Cabozantinib

Cabozantinib, initially developed against AXL e MET, demonstrated activity versus a broad range of TK, such as VEGFR2, ROS1, c-KIT, TIE2 and RET. The first report of response to cabozantinib in RET-fusion positive lung adenocarcinomas was described by Drilon and colleagues [54] in three patients as preliminary data of a phase II trial. Final results of this phase II study [43] showed 28% ORR and 73% of dose reduction rate (DRR) due to AEs in the 26 patients with RET-rearrangement NSCLC, that have been treated with cabozantinib. Similarly, in the global multicenter registry in patients with RET-rearranged lung cancers, Gautschi et al. reported 32% ORR in the 21 patients treated with cabozantinib [45].

### 3.3. Lenvatinib

Lenvatinib is a MKI of FGFRs, VEGFRs, PDGFR-alpha, KIT and RET. This MKI has been tested as oral monotherapy in 25 RET-rearranged NSCLC patients in a phase II study [62,64]; the reported ORR and DCR were 16% and 76%, respectively. In the study, almost all patients treated with lenvatinib experienced at least one treatment-related AE: grade 3 or 4 AEs was reported in 92% of patients, 16 patients (64%) required a dose modification of the therapy and six patients (24%) discontinued treatment due to side effects. In the GLORY database only two patients with RET-positive NSCLC received lenvatinib: one experienced partial response (PR) to treatment while a disease progression was reported for the second patient [45]. 

### 3.4. Other MKIs

Clinical data regarding the activity of other MKIs (sorafenib, sunitinib, ponatinib, alectinib, nitedanib and regorafenib) in RET fusion-positive NSCLC are lacking or have been reported in smaller experiences, case reports and in patients included in the large retrospective series already cited [45], in which MKIs were administered in various line of systemic therapy. The efficacy of sorafenib has been tested in a limited number of patients (*n* = 3) in a study by Horiike et al. [66]: one patient experienced stable disease (SD) while two showed progressive disease (PD) as best responses to treatment. Conversely, response to MKI sunitinib has been described in a case report of a patient with NSCLC harboring KIF5B-RET rearrangement [67]. Clinical activity of the anaplastic lymphoma kinase TKI alectinib in RET-rearranged NSCLC was firstly reported in two of four patients described by Lin et al. [60], successively in a case report [68] and in three among the four patients described in a case series [69]. The dose-limiting toxicity (DLT) to alectinib resulted from a phase I study, led to 1 level of dose-reduction recommended for ongoing phase II [70]: preliminary results from phase 2 showed 4% ORR (1 pt) and 52% DCR (13 pts) among in twenty-five RET inhibitor-naïve patients treated with 450 mg alectinib twice daily [71]. Moreover, among the 165 patients with RET-rearranged NSCLC accrued in the global retrospective registry (GLORY) [45], 53 patients (32%) were treated with RET MKIs. Of them, ten patients received sunitinib and two reported partial response (22% ORR), one of the two patients treated with nintedanib achieved a complete response, while none of the patients treated with sorafenib (two patients), alectinib (two patients), ponatinib (two patients) and regorafenib (one patient) experienced OR to these agents. Finally, conversely to other multi-kinase RET-inhibitors, RXDX-105 is a MKIs with a high potency against RET and BRAF while it is VEGFRs sparing [72]. Despite these factors, the overall activity of RXDX-105 in patients with RET fusion-positive NSCLC did not differ substantially from the activity of other MKIs. In a phase I/Ib trial [63] the reported ORR with RXDX-105, in the cohort of RET inhibitor-naïve patients with RET fusion positive NSCLC, was 19% (6/31). Interestingly, although KIF5B-RET is the most common RET fusion in NSCLC, RXDX-105 demonstrated activity only in non-KIF5B-RET- containing NSCLC. In this trial the response rate varied significantly from 0% in KIF5B-RET rearrangement NSCLCs to 67% in non- KIF5B-RET lung cancers. Interestingly, poor clinical outcomes have also been reported in patients with NSCLC harboring KIF5B-RET rearrangement treated with other above mentioned MKIs in several phase II trials [43,49,61,62], compared to patients with non-KIF5B-RET NSCLC. However, in the GLORY study, the reported clinical benefits in patients did not differ substantially based on different RET fusion identified [45] (Table 2).

## 4. New RET-Selective Inhibitors

None of the drugs described so far have been designed to preferentially bind to RET and, probably due to poor pharmacokinetic features and off-target side effects, were associated with modest clinical activity. It has been hypothesized that RET-specific antagonists could have achieved better clinical outcomes in patients harboring RET-rearranged NSCLC. Recently, two highly potent and selective RET TKIs, selpercatinib (LOXO-292) and pralsetinib (BLU-667), have been developed and their activity has been investigated in early phase trials. These agents, specifically tailored to target the activated forms of RET while sparing other kinases, offer the potential for a better clinical efficacy with a more satisfactory side effect profile. Selpercatinib has > 100-fold selectivity against VEGFR2 [75] and pralsetinib has 87-fold selectivity against VEGFR2 and 20-fold selectivity against JAK1 [74]. Furthermore, both are effective in inhibiting the RETV804L/M gatekeeper mutants and they are effective in the central nervous system [76]. Preclinically, selpercatinib (LOXO-292) demonstrated potent RET-selective antitumor activity both in in vitro and in vivo models, against both RET wild-type RET and RET alterations, with minimal activity against other kinase targets [75,77]. In a clinical setting, recent results from phase 1/2 LIBRETTO-001 trial [73] reported that selpercatinib achieved durable ORs in patients with advanced NSCLC marked by RET gene fusions. Of the first 105 enrolled patients with RET fusion-positive NSCLC, previously treated with platinum-based chemotherapy, the ORR was 64%, including two patients (2%) with a complete response and 65 patients (62%) with partial response. The median duration of response was 17.5 months and the median PFS was 16.5 months. The objective intracranial response was 91% (10/11 pts) among this cohort. Notably, in the subgroup of 39 previously untreated patients, the ORR was 85% without median PFS or OS reached at the intermediate follow-up of 9.2 months. Antitumor activity of selpercatinib was observed regardless of the specific RET fusion partner. The most common severe AEs were hypertension, hepatotoxicity, hyponatremia and lymphopenia, dose reduction was warranted in 30% of patients, but only 2% discontinued selpercatinib due to a drug-related AE. Results from LIBRETTO-001 trial led to the FDA-approval of selpercatinib for patients harboring RET-positive NSCLC, in May 2020. An ongoing phase III trial (NCT04194944) [78] is evaluating selpercatinb versus platinum-based chemotherapy (CT) with or without immunotherapy (IT) in treatment-naïve patients with advanced RET-fusion positive NSCLC. The next-generation TKI pralsetinib (BLU-667), selectively developed to target RET, demonstrated meaningful preclinical activity in a wide variety of tumors with activated RET kinase [79,80]. Preliminary data from the ongoing phase 1/2 ARROW trial (NCT03037385) demonstrated potent and durable activity and tolerability of pralsetinib in the cohort of patients with advanced RET-fusion positive NSCLC [74]. Among 116 patients with RET-positive NSCLC, 80 patients had received prior platinum treatment and 26 patients were treatment-naïve: the ORRs were 61% and 73%, respectively, with a DCR of 93% in the overall population. Furthermore, ORR was similar regardless of RET fusion partner (72% of patients had KIF5B-RET fusion NSCLC, 16% had CCDC6-RET fusion NSCLC and 12% presented other fusion) or central nervous system (CNS) involvement (56%). Most treatment-related AEs were grade 1–2 and included anemia, hepatotoxicity, constipation and hypertension. Discontinuation of treatment due to side effects was reported in 4% of patients of the safety population (all tumor types) [81]. In September 2020 pralsetinib received FDA-approval for the treatment of RET-fusion positive NSCLC patients. The international randomized phase III AcceleRET Lung study (NCT04222972) [82] is currently evaluating pralsetinib compared to standard of care as first-line in RET-positive metastatic NSCLC (Table 2).

## 5. Immune Checkpoint Inhibitors (ICI) and Chemotherapy (CT)

Only a few retrospective analyses based on limited amounts of patients have explored the effect of ICI as a single agent, suggesting a lack of benefits from this strategy. Particularly data from 4 retrospective analysis revealed similar characteristics among RET positive patients (i.e., young age, female gender, mainly non- or former smokers, adenocarcinoma histotype and low expression of Programmed Death Ligand 1 (PD-L1) with disappointing PFS (2.1 to 7.6 months) and a median OS of 12.3 months. Moreover, the median OS was not reached in patients who underwent ICI earlier in their clinical history [83,84,85,86]. A correlation between different fusions in NSCLC, gender and PD-L1 or TMB expression with a poor outcome in female that frequently express the KIF5B-RET rearrangement associated with a high rate of PD-L1 had been identified [34,83]. Moreover, the role of GDNF secreted by cells in micro-environment as stimulus in PD-L1 cell expression via JAK/STAT1 in HNSCC has been demonstrated [87]. These evidence suggest, as already well-known, the lack of benefit from ICI single agent in driven-mutation NSCLC and the lack of predictive value of PD-L1. The encouraging results from the IMpower150, a phase III trial designed to evaluate a first line combination of ICI and CT also in patients with EGFR and ALK driver alterations, open to the opportunity to explore this therapeutic option also in rearranged RET patients [88]. However, the appearance of resistance mutations to MKI and TKI, the paucity of data on treatment with ICI alone, the lack of favorable data on the immunochemotherapy combination as well as the impossibility of adequate stratification of patients who could benefit from these treatments, suggests the need to try combination therapies including targeted therapies and ICI as in some recent experiences in RET positive HCC [89]. CT alone deserves a historical mention due to its role in the last decades as a therapeutic strategy in RET rearranged NSCLC. Data from GLORY global single database about the impact of this strategy in RET fusion positive NSCLC suggested a partial efficacy of the platinum combination with pemetrexed as first line (PFS 6.3 months–OS 23.6 months) [45]. Few data in the second line, and the advent of drugs with recent FDA approval (https://www.fda.gov/drugs/resources-information-approved-drugs/fda-approves-pralsetinib-lung-cancer-ret-gene-fusions accessed on 8 January 2021) (https://www.fda.gov/drugs/drug-approvals-and-databases/fda-approves-selpercatinib-lung-and-thyroid-cancers-ret-gene-mutations-or-fusions accessed on 8 January 2021), relegate platinum based plus pemetrexed combination in subsequent lines, and in any case, after selective targeted therapies and available new drugs in clinical trials [86].

## 6. Resistance

Broadly speaking, a well-established mechanism of TKIs resistance is the development of secondary (or acquired) resistance mutations within the target kinases. These secondary somatic mutations, dynamically evolved under the selective pressure of specific TKI, enable the persistent activation of the kinases despite the presence of inhibitors. Typically, in oncogene-addicted NSCLCs, acquired mutations occur at the gatekeeper position or at the solvent front area of the kinase. These alterations confer resistance through steric interference that hinder the accessibility of drugs to the kinase ATP-binding pocket or alter the conformation of the kinase when non-contact residues are involved. Both gatekeeper and solvent front mutations have been described in different types of oncogene-driven NSCLCs. Examples of gatekeeper mutations are T790M in EGFR-mutant NSCLC, L1196M in ALK-rearranged NSCLC and L2026M in ROS-1 positive NSCLC, while typical solvent front mutations are G1202R in ALK-rearranged NSCLC and G2032R in ROS-1 positive NSCLC [90,91,92,93]. As regards RET-positive cancers, RET gatekeeper mutations at the V804 residue (V804L and V804M) primarily occur as germline mutations in sporadic medullary thyroid cancers and in about 2% of MEN2 where they act as primary driver mutations and cause intrinsic resistance to several MKIs [94]. Importantly, V804M and V804L mutations also represent the two best known secondary somatic mutations that emerge during MKIs therapies and confer resistance to treatment. Preclinically, V804M/L mutant models resulted in pan-resistance to several MKI, such as cabozantinib, vandetanib, lenvatinib and partially ponatinib, in different studies [58,80,95]. V804M/L mutations have also been reported in clinical experiences to confer resistance to vandetanib in RET-positive NSCLC patients [96,97]. 

In addition to gatekeeper mutations, solvent-front mutations at the G810 residue (G810A and G810R) and other mutations like S904F and I788N may be involved in secondary resistance. The G810A solvent-front mutation has been identified as a novel resistance mutation to vandetanib in cell lines expressing KIF5B-RET. However, although the RET G810A mutant cells conferred resistance to vandetanib, they acquired novel sensitivity to other MKIs, such as lenvatinib and ponatinib [58]. The missense S904F mutation occurs in the activation loop of the kinase domain, and is able to increase the autophosphorylation activity of RET kinase and confer resistance to vandetanib in vitro through an allosteric effect and has been also reported as a mechanism of acquired resistance in a patient with NSCLC harboring CCDC6-RET fusion after treatment with vandetanib [98]. Noteworthy, S904F mutation has also been described as a germline oncogene mutation with a high transforming activity, and implicated in the development of medullary thyroid cancer [99]. Moreover, in vitro analysis of KIF5B- or CCDC6-RET-rearranged cells identified I788N somatic mutation as a mechanism of acquired resistance to different MKIs, such as cabozantinib, vandetanib and AD80, but not to ponatinib [100]. 

Beyond the acquisition of secondary resistance mutations, another mechanism of acquired resistance in oncogene-driven NSCLC is the reactivation of different intracellular pathways, bypassing signals mediated by targeted receptor-kinase [90,101]. In RET-rearranged tumors examples of these intracellular reactivated networks include RAS/MAPK signaling, which has been reported to confer resistance to MKI AD80 in RET-rearranged cell lines and to MKI ponatinib in preclinical patient-derived models of RET-fusion positive lung adenocarcinoma [100,102]. The retained activation of EGFR and AXL signaling may contribute to the acquired resistance to MKIs, by up-regulating downstream signaling through MAPK and PI3K/AKT, respectively [102,103,104]. EGFR increases phosphorylation of RET in cell lines with neuroendocrine features and expression of Achaete-scute homolog 1 (ASCL1) suggesting a close cross-talking between both the RTKs and the interesting chance of combining different selective TKIs [105]. Also, the NRAS p.Q61K oncogenic mutation proved to represent another mechanism of acquired resistance to RET inhibition, again through MAPK and PI3K/AKT signaling reactivation [102]. Moreover, the increased Src activation has been reported as a further mechanism of acquired resistance to different MKIs by activating RET downstream effector ERK1/2 in RET-rearranged lung adenocarcinoma [106]. Finally, the MDM2 (a p53 antagonist) amplification has also been identified as a potential mediator of both intrinsic and acquired resistance to cabozantinib in patients with RET-rearranged lung cancers [107]. 

The next-generation RET-selective inhibitors selpercatinib and pralsetinib have been developed to surpass the limitations of MKIs both by sparing non-RET target kinases and by overcoming most common MKIs resistance mutations. These drugs have demonstrated equipotent and selective preclinical activity against RET rearrangements and mutations, including CCDC6-RET fusion, KIF5B-RET fusion, RET-activating mutations (C634W and M918T) and RET mutations at the gatekeeper residue (V804 L/M/E) [77,80]. Thanks to a different binding mode from MKIs, new selective RET inhibitors can avoid interference from the gatekeeper mutations, however they remain susceptible to secondary resistance from non-gatekeeper mutations. Moreover, new mechanisms of resistance have been described in preclinical models and in clinical experiences and represent an ongoing area of research. Recent studies reported that mutations at solvent front (G810R/S/C/V), hinge (Y806C/N) and β2 strand (V738A) sites within the RET kinase domain can mediate acquired resistance to selpercatinib and pralsetinib in RET fusion-positive NSCLC and in RET-mutant medullary thyroid cancer [108,109] (Table 3). A recent multi-institutional study analyzed tumor and plasma biopsies from 18 patients with RET-rearranged NSCLC after treatment with selpercatinib and pralsetinib to characterize mechanisms of acquired resistance. The analysis detected the solvent front G810C/S mutations in two cases (10%) and identified MET amplification as recurrent mechanisms of resistance (three patients, 15%), and additionally described KRAS amplification in one resistant case [110]. In addition, MET amplification has been described in a NSCLC patient in a recently published case report [111]. These evidence are in line with the knowledge that tumor cells through primary or secondary mechanisms of adaptations, overcome the inhibition with the re-activation of other signaling up- or downstream pathways. 

In addition to the importance of the rising mutation or activation of alternative pathway via amplification, the moderating role of the microenvironment should be taken in consideration [114]. For instance, in an in vitro experiment in which human umbilical vein endothelial cells (HUVECs) have been used to mimic the tumor microenvironment GDNF was able to stimulate the hepatic growth factor (HGF) production and consequently the phosphorylation of its main receptor MET [115]. As GDNF is highly expressed in NSCLC [116], we could hypothesize in RET inhibitors resistant cancer cells harboring MET amplification, a cross-talking between tumor microenvironment and TK receptors. GDNF upregulates PD-L1 involved in local immune activity and leaning immune evasion [87]. With this in mind, the importance of targeting receptors or the ligand as per GDNF and understand how to modulate the cancer microenvironment is clear. To further complicate the phenomena of resistances, attention has been recently drawn to cancer cells’ regulatory functions of miRNAs. In RET positive MTC, among several miRNAs identified, eight were up-regulated and one of them, miR-153-3p, was found to have tumor suppressor function, increasing the antiproliferative efficacy of cabozantinib by acting on factors involved in the mTOR pathway [117]. In NSCLC some experiences outline for miR-153-3p the function of a favorable prognostic factor when highly expressed, able to overcome of resistances to TKIs in mutated EGFR cell lines [118,119]. Mechanisms of regulation of endocytosis and RTKs cell trafficking are emerging news in pro-proliferative cancer cells behavior [120]. Several papers describe how alteration in RTKs degradation through endosomes and finally lysosomes, may affect cell proliferation, survival and migration [121]. Although it is intuitive that a lack of degradation of the RTKs could prolong the signaling, surprisingly also the intracellular accumulation inside the vesicles responsible for their removal, creates a feedback that increases the leading signals as per ERK1/2 or Akt in NSCLC [121,122]. In RET rearranged cell lines the expression of Golgi-Localized, Gamma Ear-Containing, ARF-Binding Protein 3 (GGA3), promote an everlasting recycling of the isoform RET51 on cell surfaces, promoting pro migratory functions via p-Akt [123] (Figure 2).

## 7. Combination Strategies and Future Perspectives

According to recent experiences, resistance mechanisms to selective RET inhibitors seem to be driven mainly by off-target RET-independent mechanisms, such as MET or KRAS amplifications [110,111] while on-target resistance mutations within RET kinases after progression to selpercatinib or pralsetinib are identified less frequently in RET aberrant NSCLC. As a consequence, new treatment approaches focus on the possibility to target RET-independent resistance drivers through the combination of anti-RET therapy with other targeted agents. Preclinical studies show that the combination with the MKI vandetanib and the mTOR inhibitor everolimus is active against CCDC6-RET-positive LC-2 lung cancer cell lines and results superior to monotherapy. Everolimus targets the PI3K/AKT pathway, reactivation of which has been associated with acquired resistance to MKIs [102]. Significant antitumor activity of everolimus plus vandetanib has been demonstrated also in patients with RET rearranged NSCLC, with responses observed in all the six patients treated within the combination therapy [124]. This combination strategy showed to increase the CNS penetration and resulted particularly active against brain-metastatic RET-rearranged NSCLCs [125]. Another promising combination therapy consists in targeting both RET activated-kinase and MET amplification. In four RET-fusion positive NSCLC patients treated with selpercatinib in the LIBRETTO-001 trial, in which MET amplification has been validated as a targetable mediator of resistance to RET-directed therapy, combined therapy with selpercatinib and the MET/ALK/ROS1 inhibitor crizotinib was administered. In this case series, the combination strategy demonstrated anecdotal evidence of clinical activity and tolerability and a 10 months-lasting response was reported with the two agents [126]. Several other combined therapies have been tested in preclinical RET altered cancer models and in early phase trials in patients with RET-positive thyroid carcinomas. Treatment with RET small interfering RNA (siRNA) and irinotecan (CPT-11) has been reported to suppress RET expression and to inhibit the growth of medullary thyroid carcinoma (MTC) xenografts via a synergic apoptotic effect [127]. The synergic antitumor activity of serine/threonine-protein kinase BRAF inhibitors (RAF256 and ZSTK474) and PI3K inhibitors (ZSTK474 and BEZ-235) on RET mediated signaling and proliferation in thyroid carcinoma cell lines harboring RET activating mutation has been described [128,129]. Combined blockade of RET and Src pathways through treatment with RPI-1 and dasatinib reduced cell proliferation in papillary thyroid carcinoma-cell lines expressing RET [130]. Moreover, association of the MKI sorafenib and a MEK inhibitor (AZD6244) demonstrated synergy in MTC cells in vitro [131], while sorafenib combined with the farnesyltransferase inhibitor tipifarnib has been evaluated in a phase I trial, showing activity in patients with RET-mutated MTC [132]. Other evidence also suggest that some repurpose drugs, such as nicotinamide, may have efficacy in RET cancer cells [133], while the use of the antibody conjugated RET-maytansine has demonstrated to be a promising strategy [134,135]. A couple of experiences in MTC and osteosarcoma cell lines has drawn the attention to the ability of piperine and ribociclib to inhibit Akt and ERK 1/2 via enhancement of activating transcription factor 4 (ATF4), suppressing RET stimuli [136,137]. In addition to combination strategies, new TKIs of different chemical scaffolds can be developed to inhibit new adaptive kinase mutants. In preclinical studies, the novel and potent RET/SRC inhibitor TPX-0046 demonstrated remarkable activity against the solvent front mutation KIF5B-RET G810R, developed as on-target resistance to selpercatinib and pralsetinib. TPX-0046 is a selective next-generation RET/SRC inhibitor, that was rationally designed with a novel macrocyclic structure and developed against various RET mutations, especially solvent front mutations [138]. BOS172738 is another novel RET inhibitor with nanomolar potency against RET and approximately 300-fold selectivity against VEGFR2 and it is currently being studied in a phase I trial (NCT03780517) [139].

Finally, further new molecules selectivity designed against RET have been tested or are currently under investigation in several preclinical trials (Table 4) [133,140,141,142,143,144]. In the past 10 years, several efforts have been made to discover highly selective small molecule RET inhibitors [145]. RET inhibitors based on heterocycles including benzimidazole, quinoline and pyrazolopyrimidine are reported in literature. These RET inhibitors are classified according to their hinge binder chemotypes as: pyrimidines, including the pyrazolopyrimidines, pyrimidine oxazines, quinazolines, 4-aminopyrimidines and 4-aminopyridines; indolinones; 5-aminopyrazole-4-carboxamides; 3-trifluoromethylanilines; imidazopyridines, imidazopyridazines and pyrazopyridines; nicotinonitriles; pyridones and 1,2,4-triazoles. Wang et al. [133] synthesized various nicotinamide analogs based on the scaffold of benzamide aminonaphthyridine HSN356, which was reported to inhibit RET kinase [142]. HSN608, the nicotinamide analog of HSN356 exerts strong RET inhibition and also inhibit RET(V804M/L) and RET(S905F) mutants better than alectinib, sorafenib, vandetanib and apatinib, and comparable to BLU667. Recently N-phenyl-7,8-dihydro-6H-pyrimido[5,4-b][1,4]oxazin-4-amine derivatives [143] have been reported as a new class of RET inhibitors and in particular 17d derivative, 1-(5-(tert-butyl)isoxazol-3-yl)-3-(4-((6,7,8,9-tetrahydropyrimido[5,4-b][1,4]oxazepin-4-yl)amino)phenyl)urea, potently inhibits RET and its drug resistance mutants RET-V804M and RET-V804L. Lakkaniga et al. [144] investigated a series of pyrrolo[2,3-d]pyrimidine-based derivatives and identified a lead compound, named 59, a type 2 inhibitor of RET, which shows low nanomolar potency against RET and RET V804M and additionally proposed a binding pose of 59 in RET pocket. Furthermore, new compounds targeting RET and VEGFR2 are emerging. The group of Moccia et al. [141] identified the clinical drug candidates Pz-1 and NPA101.3,who by lacking the structural liability for demethylation showed a selective inhibitory profile for both VEGFR2 and RET (WT and V804M).

## 8. Conclusions

In the past decades RET oncogene has emerged as a critical tumorigenesis driver. RET mutations and rearrangements now represent a well-established mechanism that drives tumor growth across several types of neoplasms, including thyroid and lung cancer. Treatment with non-specific MKIs in RET fusion-positive NSCLC achieved modest clinical outcomes and limited response durability, especially when compared with those achieved by targeting oncogenic drivers other than RET. The two highly selective RET inhibitors, pralsetinib and selpercatinib, were specifically developed to spare non-RET target kinases and to overcome resistances to MKIs. These next-generation compounds have received FDA breakthrough designation and have been approved for clinic use based on the results of the LIBRETTO-001 and ARROW trials. Although these agents have been developed to overcome MKIs limits and have demonstrated remarkable clinical activity, new mechanisms of acquired resistance have already been reported. The emergence of off-target RET-independent mechanisms of resistance to pralsetinb and selpercatinib has highlighted the necessity to test further next-generation agents and to explore new therapeutic strategies, including concurrent inhibition of RET and parallel signaling pathways of resistance.

## Figures and Tables

**Figure 1 cancers-13-01091-f001:**
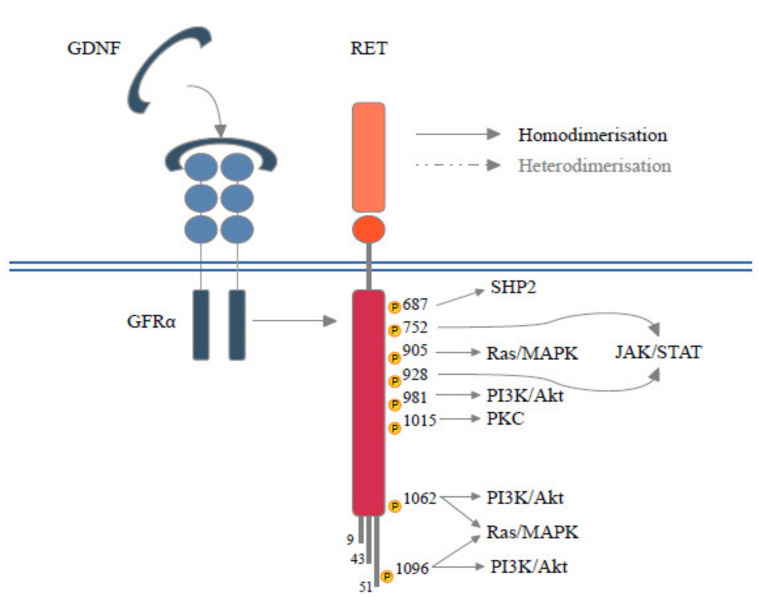
Schematic RET protein structure showing phosphorylation sites. RET forms a heterocomplex with GFRα and GFLs proteins, which in turn results in the activation of multiple signaling pathways involved in survival, differentiation, motility, proliferation, and growth.

**Figure 2 cancers-13-01091-f002:**
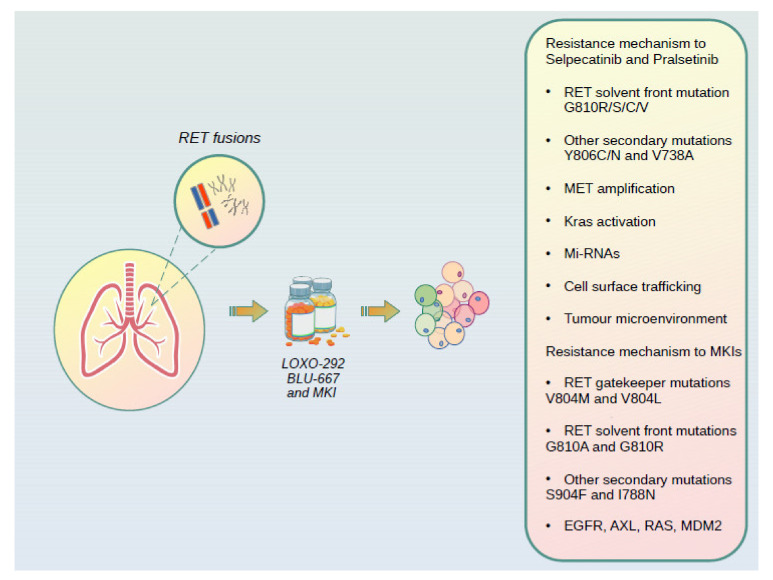
Most common mechanisms of resistance of multikinase inhibitors (MKIs) and new selective RET-inhibitors selpercatinib and pralsetinib.

**Table 1 cancers-13-01091-t001:** Other rearrangements and RET fusions.

Fusions	Histotype	Gender	Reference
CDC123-RET	ADC	F	[42]
CCDC6-RET	ADC, NE	M > F	[34]
CLIP1-RET	ADC	NA	[43]
CUX1-RET	ADC	M	[44]
EPHA5-RET	ADC	NA	[45]
ERC1-RET	ADC	NA	[43]
FRMD4A-RET	ADC	F	[46]
FYCO1-RET	ADC	F	[34]
ITGA8-RET	ADC	M	[34]
ITIH2-RET	AS	F	[34]
KIF13A-RET	ADC	F	[47]
KIF5B-RET	ADC, NE, NSCLC, AS	F > M	[34]
KIAA1468-RET	IMA	M	[48]
MIR3924-RET	SCC	M	[34]
MYO5C-RET	ADC	NA	[49]
NCOA4-RET	ADC	F	[50]
PICALM-RET	ADC	NA	[45]
RASSF4-RET	ADC	NA	[51]
RUFY2-RET	ADC	NA	[52]
SLC25A36-RET	ADC	F	[34]
SLC39A8-RET	ADC	F	[34]
TBC1D32-RET	ADC	F	[53]
TRIM24-RET	ADC	NA	[52]
TRIM33-RET	ADC	F	[54]
WAC-RET	ADC	F	[55]
ZBTB41-RET	ADC	M	[34]

Abbreviations: ADC adenocarcinoma; NE neuroendocrine; AS adenosquamous carcinoma; NSCLC non-small cell lung cancer; IMA invasive mucinous adenocarcinoma; F: female, M: male; NA not available.

**Table 2 cancers-13-01091-t002:** Antitumor activity of multikinase inhibitors (MKIs) and RET-selective inhibitors in patients with RET-positive lung cancer. Data of principal clinical trials.

Drug	Principal Kinase Targets	Type of Study	No. of pts with RET Positive—NSCLC Treated	ORR (%)	DCR (%)	DRR (%)	Grade ≥ 3 AEs (%)	Common Grade 3 or 4 TEAEs (%)
**Vandetanib**	VEGFR, EGFR, RET	Phase II trial [61]	19	47%	90%	53%	58%	Hypertension (58%) Rash acneiform (16%) Diarrhea (11%) Prolonged QTc (11%)
		Phase II trial [49]	17	18%	65%	22%	28%	Hypertension (17%) Prolonged QTc (11%) AST/ALT elevation (6%)
		Retrospective series [45]	11	18%	45%	NA	NA	NA
		Retrospective series [65]	3	0%	33%	33%	NA	NA
**Cabozantinib**	VEGFR2, MET, AXL, c-KIT, FLT3, TIE-2, RET	Phase II trial [43]	26	28%	100%	73%	47%	AST/ALT elevation (16%) Lipase elevation (15%) Decreased platelet count (8%) Hypophosphatemia (8%)
		Retrospective series [45]	21	33%	57%	NA	NA	NA
**Lenvatinib**	VEGFR1-3, FGFR1-4, PDGFR-A, c-KIT, RET	Phase II trial [62,64]	25	16%	76%	64%	92%	Hypertension (56%) Hyponatremia (20%) Proteinuria (16%) Pneumonia (16%) Nausea (12%)
		Retrospective series [45]	2	50%	50%	NA	NA	NA
**Other MKIs**								
**Sorafenib**	VEGFR1-3, PDGFRB, c-KIT, FLT3, BRAF, c-RAF	Phase II trial [66]	3	0%	33%	33%	33%	HFS (33%) Infection (33%)
		Retrospective series [45]	2	0%	100%	NA	NA	NA
**Sunitinib**	VEGFR1-3, PDGFRB, c-KIT, FLT3, RET	Retrospective series [45]	10	22%	50%	NA	NA	NA
		Case report [67]	1	-	-	-	-	Fatigue Thrombocytopenia
**Ponatinib**	BCR-ABL, FLT3, SRC, c-KIT, FGFR, VEGFR, PDGFR, RET	Retrospective series [45]	2	0%	100%	NA	NA	NA
**Alectinib**	ALK, LTK, CHEK2, FLT3, RET	Case series [60]	4	50%	75%	25%	25%	Hyperbilirubinemia (25%) Increased CPK (25%)
		Retrospective series [69]	4	50%	50%	0%	0%	None
		Phase II trial [71]	25	4%	52%	0%	4%	Neutropenia Pneumonitis Diarrhea Hyponatremia Increased CPK Hyperbilirubinemia (percentages NA)
		Retrospective series [45]	2	0%	0%	NA	NA	NA
		Case report [68]	1	-	-	-	-	None
**Nintedanib**	PDGFRA-B, VEGFR1-3, FGFR1-3	Retrospective series [45]	2	50%	100%	NA	NA	NA
**Regorafenib**	VEGFR1-3, PDGFRB, c-KIT, FGFR, RET, c-RAF	Retrospective series [45]	1	0%	0%	NA	NA	NA
**RXDX-105**	RET, BRAF	Phase I/Ib trial [63]	40	15% (19% in previous untreated pts)	52%	5%	NA	Hypophosphatemia AST/ALT elevation Rash Diarrhea Fatigue (percentages NA)
**New RET-selective inhibitors**								
**Selpercatinib**	RET	Phase I/II trial [73]	144 (105 evaluable for response)	64% (85% in previous untreated pts)	92% (95% in previous untreated pts)	30% of the safety population (3/531 pts)	28%	AST/ALT elevation (23%) Hypertension (14%) Hyponatremia (6%) Lymphopenia (6%)
**Pralsetinib**	RET	Phase I/II trial [74] *	116	61% (73% in previous untreated pts)	93%	4% of the safety population (4/120)	28%	AST elevation (22%), Hypertension (18%) ALT elevation (17%) Fatigue (15%) Neutrophilia (15%)

Abbreviations: pts, patients; NSCLC, non-small cell lung cancer; ORR, objective response rate; DCR, disease control rate; DRR, dose reduction rate; AEs, adverse events; TEAEs, treatment-emergent adverse events; AST/ALT, aspartate/alanine aminotransferases; HFS, hand-foot syndrome; CPK¸ creatinine phosphokinase; NA, not available. VEGFR, vascular endothelial growth factor receptor; EGFR, epidermal growth factor receptor; RET, rearranged during transfection proto-oncogene; MET, MET proto-oncogene, receptor tyrosine kinase; AXL, AXL receptor tyrosine kinase; c-KIT, KIT proto-oncogene receptor tyrosine kinase; FLT3, fms related tyrosine kinase 3; TIE2, tyrosine kinase with immunoglobulin-like and EGFR-like domains 2; FGFR, fibroblast growth factor receptor; PDGFRA(B), platelet derived growth factor receptor alpha (beta); BRAF, v-raf murine sarcoma viral oncogene homolog B1; c-RAF, RAF proto-oncogene serine/threonine-protein kinase; BCR-ABL, breakpoint cluster region-Abelson murine leukemia viral oncogene homolog 1; SRC, SRC proto-oncogene, non-receptor tyrosine kinase; ALK, ALK receptor tyrosine kinase; LTK, leukocyte tyrosine kinase; CHEK2, checkpoint kinase 2. * preliminary data.

**Table 3 cancers-13-01091-t003:** Mechanisms of resistance and IC_50_ (μM) for each drug.

Mutation Status		Cabozantinib [112]	Vandetanib [112]	Lenvatinib [112]	Ponatinib [112]	Selpercatinib [109]	Pralsetinib [109]
**Gatekeeper**	V804M	4.26	5.83	5.42	0.0339	0.0559	0.0168
	V804L	3.22	6.10	10.60	0.43 [60]	0.0172	0.0018
**Solvent front**	G810A	0.22	2.76	0.11	0.008 [60]	-	-
	G810R	-	-	-	-	2.744	2.650
	G810S	1.05	5.47	0.67	-	0.8802	0.3906
	G810C	-	-	-	-	1.227	0.6417
**Other**	S904F	-	0.908 [98]	-	-	-	-
	Y806C	-	0.933 [113]	-	-	0.1744	0.2958
	Y806N	4.76	5.86	1.93	-	0.1498	0.2925
	V738A	1.20	1.05	2.35	-	0.2388	0.1775

The IC_50_ values are mean (95% confidence interval). In red: resistant; in green: non-resistant. Values refers to BaF3 cell line, exception for Vandetanib Y806C value obtained in HEK 293.

**Table 4 cancers-13-01091-t004:** Candidates in preclinical setting.

Drugs	In Vitro			In Vivo	References
	RET IC_50_	VEGFR2 IC_50_	RET V804M IC_50_	Xenograft Mouse Model	
Pz-1	<0.001 μM	<0.001 μM	<0.001 μM	10 mg/kg/day per os inhibition of tumor growth	[140]
NPA-101.3	0.001 μM	0.003 μM	0.008 μM	10 mg/kg/day per os inhibition of tumor growth	[141]
HSN356	-	-	-	-	[142]
HSN608	3.16 nM	-	-	-	[133]
17d	0.01 μM	-	0.015 μM *	10–30 mg/kg/day inhibition of tumor growth	[143]
59	0.0068 μM	-	13.51 nM	-	[144]

* 0.009 μM in the V804L.

## Data Availability

Not applicable.

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
