# Peer review of "Chasing the Target: New Phenomena of Resistance to Novel Selective RET Inhibitors in Lung Cancer. Updated Evidence and Future Perspectives"

_cancers, 2021, doi:10.3390/cancers13051091_

Round 1

Reviewer 1 Report

The review manuscript is on an interesting topic in an area of very active research to identify new drugs that target the specific drivers of various cancers. Kinases inhibitors have the tendency to have significant off target adverse effects. Resistance to the kinase inhibitors through target mutations are common. The captivating title of this review article gave the impression that it will discuss the molecular/mechanistic design approaches for the kinase inhibitors aimed at resolving constant mutational changes that allow the tumors to become drug resistant. The review, though filled with useful information, appears somewhat dissociated from what the title suggests.

The language style for the manuscript leaves far too much to be desired. It is evident that the authors’ principal business language is not English. It would therefore have been appropriate to get someone who is proficient in English, and hopefully in the biomedical sciences, to help them structure the sentences better and ensure also that the right words are used. Examples include the use of “experiences” throughout the manuscript for what I suspect is “experiments”, “aware about”, “evidences”, “composed by four cadhedrin-like”, “pathways involved during”, “it has recently been recognized a role of RET”, “the mainly involved gene in in fusion”, “by incrementing ATP binding”, “activity have”, “target therapies”, “amplification have been”, “resulted effective against”, “those obtained targeting etc. These are only a tiny portion of the grammatical errors and other language issues that make it extremely difficult to follow the manuscript. Furthermore, in vivo and in vitro should be italicized, and 50 in IC50 should be subscripted.

Could section “2.1. Germline mutations” be summarized in the form of a table? Could the kinase mutations and drugs targeting them be organized in the form of a table in drug development chronological terms? It would even be better if the design elements of the drugs that make them more effective against the mutant proteins are briefly mentioned.

How is section 5 related to RET? More specifically, how does it fit into the search for agents that target RET mutations?

What does “…we could hypothesize in RET inhibitors resistant cancer cells harboring MET amplification,…” mean?

How is the discussion different from the preceding sections? What is being discussed when no new data is in the manuscript?

Author Response

Reviewer 1 comments:

The review manuscript is on an interesting topic in an area of very active research to identify new drugs that target the specific drivers of various cancers. Kinases inhibitors have the tendency to have significant off target adverse effects. Resistance to the kinase inhibitors through target mutations are common. The captivating title of this review article gave the impression that it will discuss the molecular/mechanistic design approaches for the kinase inhibitors aimed at resolving constant mutational changes that allow the tumors to become drug resistant. The review, though filled with useful information, appears somewhat dissociated from what the title suggests.

We agree with the criticism highlighted by the reviewer and accordingly we have edited the title in the revised version of the manuscript: in the title we added “in lung cancer” to clarify that the manuscript is focused on RET-positive lung cancer. We changed “evidences” with “evidence”, as suggested by reviewer 1.

The language style for the manuscript leaves far too much to be desired. It is evident that the authors’ principal business language is not English. It would therefore have been appropriate to get someone who is proficient in English, and hopefully in the biomedical sciences, to help them structure the sentences better and ensure also that the right words are used. Examples include the use of “experiences” throughout the manuscript for what I suspect is “experiments”, “aware about”, “evidences”, “composed by four cadhedrin-like”, “pathways involved during”, “it has recently been recognized a role of RET”, “the mainly involved gene in in fusion”, “by incrementing ATP binding”, “activity have”, “target therapies”, “amplification have been”, “resulted effective against”, “those obtained targeting etc. These are only a tiny portion of the grammatical errors and other language issues that make it extremely difficult to follow the manuscript.

We regret there were problems with the English. The paper has been carefully revised by a native English speaker to improve the grammar and readability.

Furthermore, in vivo and in vitro should be italicized, and 50 in IC50 should be subscripted.

We have revised the text according to reviewer suggestions: “in vivo” and “in vitro”  have been italicized (lines in the revised version as follow: 170, 285, 405, 409, 463, 535) and 50 in “IC50” has been subscripted, as requested.

Could section “2.1. Germline mutations” be summarized in the form of a table?

Thank you for your kind suggestion. We have edited the title in the revised version of the manuscript to clarify that the manuscript is focused on RET-positive lung cancer. Since germline mutations are not involved in lung cancer, we decided not to include a specific table to summarize section 2.1 “Germline mutation”.

Could the kinase mutations and drugs targeting them be organized in the form of a table in drug development chronological terms? It would even be better if the design elements of the drugs that make them more effective against the mutant proteins are briefly mentioned.

We are grateful to the reviewer for suggesting the inclusion of a new table to improve the readability of the manuscript. We added Table 3 where are described the most common kinase resistance mutations in RET-positive lung cancer cell lines and the IC50 values of drugs, available from literature. The first column of the table shows data concerning MKIs, while the two recently approved TKIs are reported in the two last columns. Although the table is not organized in a punctual chronological order, we added two shades of color to quickly visualize resistant/not-resistant mutations for each drug. Consequently, herein we added two new references (Liu X, British J of Pharmacology 2018 [ref 115]; Carlomagno, Endocr Relat Cancer 2009 [ref 116]).

As asked by the reviewer regarding the design elements, we better explained in the revised version of the manuscript (from line 554 to line 579 in the revised version of the manuscript). In this section the design elements of selective RET inhibitors, including new molecules described in Table 4, are briefly pointed out. A new reference was added for this section (Cong-Cong Jia, Fut Med Chem 2020 [ref 148]).

How is section 5 related to RET? More specifically, how does it fit into the search for agents that target RET mutations?

Based on phase III trial ImPower150, EGFR mutated and ALK translocated lung cancer patients seem to benefit from the combination of chemotherapy with immunotherapy, suggesting that the same scenario can be  achievable in RET-positive lung cancer. In other cancer types there are effective combinations between TKI and immunotherapy (see [ref 98] about RET-positive HCC).  In paragraph 5 we have briefly explored the actual knowledge and data that refer to chemotherapy and immunotherapy and eventually their failure as single agents.

What does “…we could hypothesize in RET inhibitors resistant cancer cells harboring MET amplification,…” mean?

We regret the statement "we could hypothesize in RET inhibitors resistant cancer cells harboring MET amplification […]" was not clear. We agree that the meaning of this argument could be misunderstood, consequently we have explained it better in the revised version of the manuscript (lines 463-671). Two references have also been cited to better support the argument (Lin, OncoImmunology 2017 [ref 90] and Mulligan, Front Physiol 2019 [ref 117]).

How is the discussion different from the preceding sections? What is being discussed when no new data is in the manuscript?

We are grateful to the reviewer for drawing our attention to this problem. Since in the section 7 we pointed out the possibility of targeting RET-independent resistance drivers through combination therapies and we described possible future approaches with new molecules, we have changed the title from “discussion” to “combination strategies and future perspectives”.

Reviewer 2 Report

It is a well-written review article of RET inhibitors.

Some concerns were raised in order to improve the manuscript.

  1. Page 1. Line 36. NSCLC -> non-small cell lung cancer (NSCLC)
  2. Table 1. MU (mucinous) -> “MUAD, means “mucinous adenocarcinoma” instead of “mucinous”.
  3. Table 1. CDC6 --> CCDC6 ; By ref 34, “males [0.9% (8/902)]. The most common fusions were KIF5B–RET in females [80% (12/15)] and CCDC6–RET in males [50% (4/8)] “ indicated that CDCC6, M=F (gender) or M>F and the histotypes were 3 AD and one NE (NE -> neuroendocrine tumor) . In addition, KIF5B, histology types were “AD, NE and NSCLC”, not SCC. Authors had to check the reference data and fusion gene carefully.
  4. it would be more readable if authors could summarize the current drugs in RET-positive NSCLC, included MKIs (Vandetanib, Cabozantinib, Lenvatinib) and New RET-selective inhibitors (lpercatinib (LOXO-292) and pralsetinib (BLU-667), included mechanism, trials, ORR, DCR, DRR (dose reduction rate) and ADR (adverse drug reaction), in one or two tables.
  5. Please summary the resistant mechanism of RET-positive patients to MKIs and new RET-selective inhibitors in a table, it would be more readable.

Author Response

Reviewer 2 comments:

It is a well-written review article of RET inhibitors.

Some concerns were raised in order to improve the manuscript.

  1. Page 1. Line 36. NSCLC -> non-small cell lung cancer (NSCLC).

 “NSCLC” abbreviation has been defined in parentheses, as requested.

  1. Table 1. MU (mucinous) -> “MUAD, means “mucinous adenocarcinoma” instead of “mucinous”.

“MU (mucinous)” has been changed with “IMA (invasive mucinous adenocarcinoma)” instead of “MUAD” as suggested by the reviewer, as it is reported in this form by the cited [ref 48].

  1. Table 1. CDC6 --> CCDC6 ; By ref 34, “males [0.9% (8/902)]. The most common fusions were KIF5B–RET in females [80% (12/15)] and CCDC6–RET in males [50% (4/8)] “ indicated that CDCC6, M=F (gender) or M>F and the histotypes were 3 AD and one NE (NE -> neuroendocrine tumor). In addition, KIF5B, histology types were “AD, NE and NSCLC”, not SCC. Authors had to check the reference data and fusion gene carefully.

  1. CDC6 -> CCDC6: typing error, revised as requested.
  2. Line 176: “NE” abbreviation has been defined as requested: “NE neuroendocrine”. We added “NSCLC non-small cell lung cancer” in the legend, as it has added also in Table 1.
  • We are grateful to the reviewer for highlighting our misunderstanding of [ref 34] (Qiu Z, Sci Reports 2020). We carefully checked [ref 34] as suggested. In this research article, the authors described the genetic and prognostic characteristics of patients with RET-fusion positive lung cancer. According to this paper, CCDC6-RET fusion was found in 6 patients, including 4 males 2 females (M>F as reported in table 1). Regarding KIF5B-RET fusion we specified all the histology types reported by [ref 34]: ADC adenocarcinoma (11 pts), NE neuroendocrine (1 pt), AS adenosquamous (1 pt) and NSCLC (1 pt). SCC does not refer to this specific RET fusion.

  1. it would be more readable if authors could summarize the current drugs in RET-positive NSCLC, included MKIs (Vandetanib, Cabozantinib, Lenvatinib) and New RET-selective inhibitors (lpercatinib (LOXO-292) and pralsetinib (BLU-667), included mechanism, trials, ORR, DCR, DRR (dose reduction rate) and ADR (adverse drug reaction), in one or two tables.

We have summarized the current drugs in RET-positive NSCLC (including MKIs and new RET-selective inhibitors), their characteristics, the trials in which each drug was tested and the most relevant clinical outcomes in Table 2, as suggested by the reviewer.

  1. Please summary the resistant mechanism of RET-positive patients to MKIs and new RET-selective inhibitors in a table, it would be more readable.

We are extremely grateful to the reviewer for pointing out this problem. To improve the readability of our manuscript, as suggested by the reviewer we have added Table 3 in the revised version of the manuscript.

Round 2

Reviewer 1 Report

The authors state that the manuscript has been thoroughly edited by someone who is proficient in the English language. This hard to believe as it is still full of errors that need correction. For example, “…inhibitors active also on RET,…” on line 127, …”…2/3 of case rise up…” on line 132, “In NSCLC the mainly involved gene in fusions is KIF5B,…” lines 145-146, “…transmembrane dominion…” line 154, “…and gender have been founded.” Line 156. These are just highlights from a small section of one page, page 4. Proper punctuation would be most helpful in understanding the manuscript.

What does “…positive estrogens receptors,…” mean? While a scientific correction is needed for this phrase, “…estrogens receptors” again does not pass the language test. The authors should seriously consider the language issues if they want the manuscript to be published.

Author Response

The authors state that the manuscript has been thoroughly edited by someone who is proficient in the English language. This hard to believe as it is still full of errors that need correction. For example, “…inhibitors active also on RET,…” on line 127, …”…2/3 of case rise up…” on line 132, “In NSCLC the mainly involved gene in fusions is KIF5B,…” lines 145-146, “…transmembrane dominion…” line 154, “…and gender have been founded.” Line 156. These are just highlights from a small section of one page, page 4. Proper punctuation would be most helpful in understanding the manuscript.

What does “…positive estrogens receptors,…” mean? While a scientific correction is needed for this phrase, “…estrogens receptors” again does not pass the language test. The authors should seriously consider the language issues if they want the manuscript to be published.

We regret there were problems with the English. The paper has been revised by a native English speaker to improve the grammar and readability and all revisions on the new version of the manuscript are highlighted in yellow.

Round 3

Reviewer 1 Report

There are still some minor errors and omissions. The authors can either correct them and/or the editorial office can help rectify them.